# Broussochalcone A Is a Novel Inhibitor of the Orphan Nuclear Receptor NR4A1 and Induces Apoptosis in Pancreatic Cancer Cells

**DOI:** 10.3390/molecules26082316

**Published:** 2021-04-16

**Authors:** Hyo-Seon Lee, Soo-Hyun Kim, Bo-Mi Kim, Stephen Safe, Syng-Ook Lee

**Affiliations:** 1Department of Food Science and Technology, Keimyung University, Daegu 42601, Korea; hing1035@naver.com; 2National Institute for Korean Medicine Development, Gyeongsan 38540, Korea; beluga81@nikom.or.kr (S.-H.K.); bom0203@nikom.or.kr (B.-M.K.); 3Department of Veterinary Physiology and Pharmacology, Texas A&M University, College Station, TX 77843-4466, USA; SSAFE@cvm.tamu.edu

**Keywords:** apoptosis, broussochalcone A, ER stress, NR4A1, pancreatic cancer, Sp1

## Abstract

The orphan nuclear receptor 4A1 (NR4A1) is overexpressed in pancreatic cancer and exhibits pro-oncogenic activity, and NR4A1 silencing and treatment with its inactivators has been shown to inhibit pancreatic cancer cells and tumor growth. In this study, we identified broussochalcone A (BCA) as a new NR4A1 inhibitor and demonstrated that BCA inhibits cell growth partly by inducing NR4A1-mediated apoptotic pathways in human pancreatic cancer cells. BCA downregulated specificity protein 1 (Sp1)-mediated expression of an anti-apoptotic protein, survivin, and activated the endoplasmic reticulum (ER) stress-mediated apoptotic pathway. These results suggest that NR4A1 inactivation contributes to the anticancer effects of BCA, and that BCA represents a potential anticancer agent targeting NR4A1 that is overexpressed in many types of human cancers.

## 1. Introduction

The orphan nuclear receptor 4A1 (NR4A1), also known as Nur77 or TR3, is activated by various stimuli and plays important roles in the central nervous system, inflammation, and metabolic processes. Previous studies have shown that NR4A1 is highly expressed in various types of human tumors, including pancreatic tumor, and plays an important role in tumor growth and metastasis [1,2,3,4].

Several anticancer agents, such as cytosporone B and its analogs, are known to induce apoptosis via NR4A1-dependent pathways in cancer cells. One of the main mechanisms involves translocation of nuclear NR4A1 to the mitochondria where it binds to bcl-2 and changes its fate from an anti-apoptotic to a pro-apoptotic protein that then activates the extrinsic apoptosis pathway [5]. Located in the nucleus, NR4A1 also plays a vital role in cell survival and apoptosis in a variety of cancer cells and tumors [1,2,3,4,6]. Studies have discovered several nuclear NR4A1-dependent pro-apoptotic pathways in cancer cells [2,3,6]. These mechanistic studies revealed that the anti-apoptotic activity of NR4A1 was linked to the formation of a specificity protein 1 (Sp1)/p300/NR4A1 complex that bound to the guanine-cytosine (GC)-rich promoter regions of several pro-survival and anti-apoptotic genes and to the downregulation of endoplasmic reticulum (ER) stress by decreasing cellular reactive oxygen species (ROS) levels [2,3].

In addition, in human pancreatic cancer cells that overexpress NR4A1, treatment with antagonists or inactivators of nuclear NR4A1, such as 1,1-bis(3′-indolyl)-1-(p-hydroxyphenyl)methane (DIM-C-pPhOH) and fanchinoline (FCN) inhibited cell growth and induced apoptosis, and this was accompanied by the downregulation of Sp1-dependent anti-apoptotic genes and induction of ROS-mediated ER stress [2,3,7]. Therefore, compounds that inactivate nuclear NR4A1 have been characterized as a new class of chemotherapeutic agents for the treatment of NR4A1 overexpression in pancreatic cancer. In addition, several synthetic and natural inactivators of nuclear NR4A1 have recently been discovered [2,3,7].

Natural products and plant-derived small molecules are widely studied as complementary and alternative medicines for treating various types of cancer, including pancreatic cancer, either alone or in combination with other existing anticancer drugs [8]. For example, chalcone derivatives, such as chalcone hybrids composed of hydroxamic acids or 2-aminobenzamide groups, have been shown to inhibit histone deacetylases, known as promising targets for cancer treatment in different human cancer cells [9].

We have long been interested in discovering naturally occurring small molecules that inactivate nuclear NR4A1 in the treatment of cancers that overexpress this orphan receptor, and we recently isolated and identified FCN, a bisbenzyltetrahydroisoquinoline alkaloid from *Stephania tetrandra*, as the first natural inactivator of nuclear NR4A1 that induces apoptosis in pancreatic cancer cells partly via nuclear NR4A1-dependent pro-apoptotic pathways as mentioned above [7]. In this study, we identified a new NR4A1 inactivator, broussochalcone A (BCA), through screening a herbal medicine-derived small molecule library, which showed that BCA induces NR4A1-dependent apoptosis in human pancreatic cancer cells.

## 2. Results and Discussion

### 2.1. BCA Inhibits NR4A1-Mediated Transactivation in Pancreatic Cancer Cells

NR4A1 is highly expressed in various human cancer cell lines and tumors from the pancreatic, breast, colon, lung, and kidney cancer patients [1,2,3,6,10]. Increasing evidence suggests that NR4A1 acts as a pro-oncogenic factor in many human cancer cells and tumors [1,2,3,4]. Earlier studies on the pro-oncogenic roles of NR4A1 did not reveal a detailed mechanism but showed that NR4A1 gene silencing inhibited cell proliferation and induced apoptosis in various cancer cells [11]. Through a series of subsequent studies, we identified and reported two novel mechanisms of NR4A1-mediated apoptosis in cancer cells [2,4]. Therefore, targeting NR4A1-mediated pro-oncogenic pathways represents an attractive therapeutic strategy for cancer overexpressing NR4A1 and there is a growing interest in the development of NR4A1 inhibitors.

In the present study, over 600 herbal medicine-derived small molecules were tested for their ability to inhibit nuclear NR4A1 using a luciferase reporter gene driven by NR4A1-binding response elements (NBRE-Luc). We found that BCA significantly inhibited NBRE reporter gene activity in Panc-1 and MiaPaCa-2 pancreatic cancer cells (Figure 1A). The inhibitory effect of BCA on NR4A1 transactivity was further investigated using a Gal4-NR4A1 fusion construct (Gal4-NR4A1) that activates a Gal4 reporter construct harboring tandem repeats of Gal4 response elements upstream of a luciferase reporter (Gal4-RE-Luc). Luciferase activity in cells transfected with Gal4-NR4A1 and Gal4-RE-Luc was highly increased in both cell lines, but treatment with BCA (10–30 µM) markedly decreased luciferase activity (Figure 1B). Moreover, Western blotting results showed that BCA treatment had no effect on the expression of the NR4A1 protein (Figure 1C), indicating that the inhibitory effect of BCA on NR4A1-mediated transactivation was not due to a decrease in protein expression of NR4A1.

NR4A1 contains two ligand-independent activation domains, activation function-1 (AF-1) and AF-2 at the N- and C-terminal regions, respectively. AF-1 is reported to be the major transactivation domain of NR4A1 in the Gal4-NR4A1 constructs, whereas AF-2 affects minimal NR4A1 transactivation [3,7]. In this study, treatment with BCA (10–30 µM) significantly inhibited luciferase activity in MiaPaCa-2 cells transfected with Gal4-RE-Luc/C-terminal deletion Gal4-NR4A1 chimera (Gal4-NR4A1-AB) and Gal4-RE-Luc (Appendix A). However, the basal luciferase activity was low in the cells transfected with the N-terminal deletion Gal4-NR4A1 chimera (Gal4-NR4A1-CF) and the Gal4-RE-Luc, which is consistent with data from previous reports [2,7], and BCA did not affect luciferase activity (data not shown), suggesting that BCA decreased the transactivity of NR4A1 specifically via the N-terminal region of NR4A1 which contains AF-1.

As the intracellular localization of NR4A1 is known to be an important factor determining its role in cancer cells, the effect of BCA treatment on the localization was evaluated by immunostaining with an anti- NR4A1 antibody. In MiaPaCa-2 cells treated with dimethyl sulfoxide (DMSO; control) for 3 h, NR4A1 was found to be predominantly located in the nucleus, whereas in cells treated with BCA (20 µM) for 3 h, it was observed in both the nucleus and cytoplasm (Figure 1D). Therefore, to determine whether the BCA-mediated inhibition of NR4A1 transactivity (Figure 1A,B) was associated with the nuclear export of NR4A1, the effect of BCA on Nur77 binding response element (NBRE) reporter gene activity was further examined in cells cotreated with BCA and the nuclear export inhibitor leptomycin B (LMB). It was first confirmed that treatment with LMB (8 ng/mL) completely blocked the BCA-induced nuclear export of NR4A1 in MiaPaCa-2 cells (Figure 2A). BCA was found to significantly inhibit the luciferase activity of NBRE-Luc in cells cotreated with LMB at a level similar to that of BCA treatment alone (Figure 2B), suggesting that BCA not only induces the nuclear export of NR4A1 but also inhibits the transactivation of nuclear NR4A1. This mechanism of NR4A1 inhibition is unique and has not been reported previously. We are currently investigating the molecular mechanisms associated with the BCA-induced nuclear export of NR4A1 in cancer cell growth and survival.

### 2.2. Inactivation of NR4A1 Contributes to Growth Inhibition by BCA in Pancreatic Cancer Cells

We previously showed that treatment with NR4A1 inhibitors such as FCN and DIM-C-pPhOH and the silencing of NR4A1 reduced cell proliferation and induced apoptosis in various cancer cells and solid tumors [1,2,3,4,7]. Therefore, in the present study, we examined the effect of BCA on cell viability and apoptosis in pancreatic cancer cells. Treatment for 24 and 48 h with 10, 20, and 30 µM BCA, but 5 µM BCA significantly inhibited proliferation of the two pancreatic cancer cells (Panc-1 and MiaPaCa-2) with growth-inhibitory IC_50_ (the half maximal inhibitory concentration) values of 21.10 µM and 27.20 µM, respectively, at 48 h (Figure 3A). In addition, BCA treatment of other cancer cells that expressed high levels of NR4A1, including AGS (human gastric adenocarcinoma cells), HCT116 (human colorectal carcinoma cells), and H460 (large cell lung cancer cells) confirmed that BCA inhibited cell proliferation in several cancer cell lines (Appendix A).

To further confirm that the BCA-mediated growth inhibition was due to inactivation of NR4A1, rescue experiments were performed using a plasmid construct expressing NR4A1 (Flag-NR4A1). As shown in Figure 3B, the overexpression of NR4A1 partially, but significantly (*p* < 0.05), rescued the growth inhibition by BCA in both MiaPaCa-2 and Panc-1 cells, indicating that inactivation of NR4A1 partly contributes to BCA-mediated growth inhibition of human pancreatic cancer cells.

### 2.3. BCA Induces NR4A1-Dependent Apoptosis in Pancreatic Cancer Cells: Sp1/Survivin-Mediated Apoptosis

Several previous studies using NR4A1 siRNA (siNR4A1) and NR4A1 inactivators revealed that NR4A1-dependent apoptosis is mediated partly by decreased expression of survivin, the anti-apoptotic gene and a potential target for human pancreatic cancer therapy [12,13]. Survivin expression in human pancreatic cancer cells is known to be modulated mainly by the oncogenic transcription factor, Sp1, that is also one of the negative prognostic factors for pancreatic cancer patients [2].

Western blotting results showed that BCA increased cleavage of PARP and caspase-8 (Figure 4A) in Panc-1 and MiaPaCa-2 cells, indicating that BCA induced apoptosis in these cancer cells. These effects of BCA were similar to those observed following transfection with siNR4A1 or treatment with the NR4A1 inactivators, FCN and DIM-C-pPhOH [2,7]. Furthermore, BCA downregulated the protein levels of survivin in both cell lines (Figure 3A), and this was comparable to the effect of BCA on their mRNA expression (Figure 4B). To further examine the effect of BCA on Sp1-dependent regulation of survivin in panc-1 cells that showed substantial decrease in survivin expression following BCA treatment, we used two survivin promoter (SVV) reporter constructs (pGL3-SVV(−150) and pGL3-SVV(−269)), and a promoter construct harboring tandem repeats of GC-rich sequences ([GC]_3_-Luc). As shown in Figure 4C,D, treatment with BCA (10–30 µM) significantly reduced luciferase activity in Panc-1 cells transfected with two surviving promoter constructs that contained 4 and 8 GC-rich regions, respectively. BCA also reduced luciferase activity in cells transfected with [GC]_3_-Luc. Our previous studies [2,7] showed that survivin gene expression was due to p300/NR4A1-mediated coactivation of Sp1 in pancreatic cancer cells, and decreased expression of survivin following treatment with FCN and DIM-C-pPhOH was associated with NR4A1 inactivation. These results suggest that the BCA-mediated decrease in survivin expression is due, at least in part, to inactivation of NR4A1, confirming the similar effects of BCA and other inactivators of nuclear NR4A1 on survivin expression [2,3,4,7].

### 2.4. BCA Induces NR4A1-Dependent Apoptosis in Pancreatic Cancer Cells: ROS/ER Stress-Mediated Apoptosis

Nuclear NR4A1 plays a role in maintaining intracellular oxidative stress at safe levels and minimizing ROS-dependent ER stress in pancreatic cancer cells by modulating the expression of thioredoxin domain-containing 5 (TXNDC5), which regulates cellular ROS production [4]. Previous studies have also shown that treatment with NR4A1 inactivators and knockdown of NR4A1-induced ROS–ER stress-dependent apoptosis through downregulation of TXNDC5 expression. Treatment with BCA (10–30 µM) decreased protein expression of TXNDC5 and increased the expression of ER stress markers, including CCAAT/enhancer-binding protein (C/EBP) homologous protein (CHOP) and glucose-regulated protein 78 (GRP78), in both the pancreatic cancer cells (Figure 5A). As expected, BCA treatment dramatically increased cellular ROS production in Panc-1 cells attenuated by cotreatment with N-acetylcysteine (NAC) (Figure 5B). Moreover, Western blot results showed that BCA-induced apoptosis and ER stress were partially attenuated in Panc-1 cells cotreated with NAC (Figure 5C), suggesting that induction of ROS-mediated ER stress also contributes, in part, to BCA-mediated apoptosis in pancreatic cancer cells.

Broussochalcone A (BCA) is a prenylated flavonoid found in the roots of *B. papyrifera*, a medicinal plant commonly found in Asian countries such as Korea, Thailand, China, India, and Japan [14]. The pharmacological activity of BCA is relatively less known compared to that of other small molecules of natural origin. Thus far, a limited number of its activities, such as antiplatelet [15], antivirus [16], and anti-inflammatory [17] have been reported. The anticancer activities of BCA have been demonstrated in several human cancer cells, including the liver [18,19] and colon [19]. The BCA-dependent anticancer mechanisms proposed in those reports include beta-catenin degradation and forkhead box protein O3 (FOXO3)-mediated apoptosis. In this study, we identified, for the first time, a new anticancer mechanism of BCA involving NR4A1-dependent apoptosis. In addition, using Gal4-NR4A1–Gal4-RE-Luc and NBRE-Luc constructs, we also identified BCA as an inactivator that targets nuclear NR4A1-mediated transactivation (Figure 1A,B). However, the inhibitory effect of BCA on the proliferation of pancreatic cancer cells was not fully restored by the overexpression of NR4A1 (Figure 3B), indicating that apart from its inhibitory effect on NR4A1-dependent transactivation, the anticancer activity of BCA may involve other mechanisms such as antagonism of the Wnt/β-catenin pathway and cell-cycle arrest, as previously reported [18,19].

In conclusion, as illustrated in Figure 5D, BCA, a natural inactivator of NR4A1, inhibited cell growth and induced apoptosis partly through NR4A1-dependent apoptotic pathways in pancreatic cancer cells. This study proposes a novel anticancer mechanism for BCA and suggests that it represents a potential treatment option for tumors that overexpress NR4A1. Currently, the functional role of BCA-induced translocation of nuclear NR4A1 in cell growth and survival of pancreatic cancer cells is being investigated.

## 3. Materials and Methods

### 3.1. Isolation of BCA from Broussonetia Papyrifera Extract

The roots of *B. papyrifera* (harvested in Gimje, Jeollabuk-do, Korea) were purchased from a common market in Gyeongsan, Gyeongbuk, Korea. The roots (5 kg) were extracted three times with 95% methanol (15 L) for 24 h at room temperature. After filtration, the crude extract was concentrated on a rotary evaporator and dried under vacuum at 40 °C. The dried extract (278.7 g) was suspended in distilled water (1000 mL) and fractionated with ethyl acetate (EtOAc). The EtOAc-soluble fraction (152.0 g) was separated into 14 fractions (fractions 1–14) on a silica gel column (Kieselgel 60, Merck, Darmstadt, Germany) using hexane and EtOAc (Hexane–EtOAc = 100:1 to 50:50). Fraction 10 (20.0 g) was further separated with a Sephadex LH-20 column (Amersham Biosciences Inc., Tokyo, Japan) using methanol to obtain four subfractions (subfractions 10-1–10-4). Among them, subfraction 10-3 (9 mg) was further purified with C_18_ reversed-phase column chromatography (ODS-A, YMC, Kyoto, Japan) by gradient elution with water/acetonitrile containing trifluoroacetic acid (TFA). High performance liquid chromatography (HPLC) was performed with Agilent 1260 series as described in Appendix A (Agilent Technologies, Santa Clara, CA, USA). The compound (800 mg) was identified as BCA by using different spectroscopic techniques, including ultraviolet (UV) absorption, 1D nuclear magnetic resonance (NMR) (^1^H, ^13^C NMR), 2D NMR (^1^H-^1^H-correlation spectroscopy (COSY), heteronuclear single quantum coherence (HSQC), heteronuclear multiple bond correlation (HMBC)), and electron ionization-mass spectrometry (EI-MS). The ^1^H and ^13^C NMR spectra were recorded on a Jeol ECA-500 MHz NMR instrument (Jeol, Tokyo, Japan) operating at 500 MHz for ^1^H NMR and 100 MHz for ^13^C NMR using tetramethylsilane (TMS) as an internal standard. EI-MS was measured using a Waters Q-TOF micro mass spectrometer (Waters, Milford, MA, USA).

The UV-vis absorption, HPLC spectrum and chemical structure of BCA (purity 99.8%; melting point 183–184 °C) isolated from *B. papyrifera* root were shown in Appendix A. Proton and carbon NMR signals were observed as follows: yellow powder, EI-MS *m/z* 340 [M]^+^, molecular formula C_20_H_20_O_5_; ^13^C NMR (Acetone-*d_6_*, 125 MHz) δ 192.7 (C=O), 165.9 (C-4′), 163.4 (C-2′), 149.1 (C-4), 146.4 (C-3), 145.3 (C-β), 132.3 (C-6′, C-9′), 128.3 (C-1), 124.1 (C-8′), 123.4 (C-6), 121.4 (C-5′), 118.5 (C-α), 116.4 (C-2), 115.8 (C-5), 114.3 (C-1′), 103.4 (C-3′), 25.9 (C-10′), 17.9 (C-11′); ^1^H-NMR (Acetone-*d_6_*, 500 MHz) δ 7.96 (1H, s, H-6), 7.75 (1H, dd, *J* = 15, H-β), 7.69 (1H, d, *J* = 15, H-α), 7.31 (1H, d, *J* = 2, H-2), 7.23 (1H, dd, *J* = 2 and 8 Hz, H-6), 6.91 (1H, d, *J* = 8.0 Hz, H-5), 6.40 (1H, d, *J* = 1.5 Hz, H-3′), 5.34 (1H, br t, *J* = 7.2 Hz, H-8′), 3.30 (2H, br d, *J* = 7.5 Hz, H-7), 1.72 (6H, s).

### 3.2. Cell Lines and Plasmids

All human cancer cell lines including MiaPaCa-2 and Panc-1 human pancreatic cancer cells were obtained from Korean Cell Line Bank (Seoul, Korea) and grown in Dulbecco’s Modified Eagle Medium supplemented with 10% fetal bovine serum (Welgene, Gyeongbuk, Korea). Cells were maintained at 37 °C in a humidified CO_2_ incubator (5% CO_2_). The Gal4–NR4A1 chimeras Gal4-NR4A1 (amino acid 1 to 598), Gal4–NR4A1–AB (amino acid 1 to 266), and Gal4–NR4A1–CF (amino acid 267–598) were constructed by inserting polymerase chain reaction (PCR)-amplified fragments into the EcoRI/HindIII site of pM vector (Clontech, Mountain View, CA, USA). All other reporter constructs have been previously described [3,6].

### 3.3. Antibodies, Reagents, Quantitative Real-Time PCR, and Western Blot Analysis

NR4A1 and thioredoxin domain-containing 5 (TXNDC5) antibodies were purchased from Abcam (Cambridge, MA, USA) and GeneTex (Irvine, CA, USA), respectively. Activating transcription factor 4 (ATF4) and glucose regulatory protein 78 (GRP78) antibodies were purchased from Santa Cruz Biotechnology (Santa Cruz, CA, USA), and all remaining antibodies were purchased from Cell Signaling Technology (Beverly, MA, USA). Reporter lysis buffer, luciferase reagent, and β-galactosidase (β-Gal) reagent were obtained from Promega (Madison, WI, USA). Quantitative real-time PCR and Western blot analysis were undertaken as previously described [3]. The sequences of the primers used for real-time PCR were as follows: survivin sense 5′-CAG ATT TGA ATC GCG GGA CCC-3′, antisense 5′-CCA AGT CTG GCT CGT TCT CAG-3′ and TBP sense 5′-TGC ACA GGA GCC AAG AGT GAA-3′, antisense 5′-CAC ATC ACA GCT CCC CAC CA-3′.

### 3.4. Cell Proliferation Assay, Transfection, and Reporter Gene Assay

Cell viability was determined by an 3-(4,5-dimethylthiazol-2-yl)-2,5-diphenyltetrazolium bromide (MTT) assay as described previously [20]. For the reporter gene assay, cells (3 × 10^4^ cells/well) were plated in 48-well plate and grown overnight. Cells were then cotransfected with 25 ng of each luciferase reporter construct and 5 ng of the pCMV-β-galactosidase reporter construct for 5 h using Lipofectamine 2000 reagent (Invitrogen, San Diego, CA, USA). After transfection, the cells were incubated with each compound for 18 h. Luciferase and β-galactosidase activities were assayed by using the luciferase and β-galactosidase enzyme assay systems, and luciferase activity was normalized with the β-galactosidase activity.

### 3.5. Subcellular Localization Assays

Cells (2 × 10^5^ cells/well) were plated onto coverslips (12 mm) in 12-well plate and treated with each compound for 3 h. The cells were then fixed in 1% formalin in phosphate-buffered saline (PBS; pH 7.4) after washing with PBS and permeabilized by immersing the cells in 0.3% Triton X-100 solution in PBS for 10 min. Cells were then incubated with anti-NR4A1 rabbit IgG, followed by anti-rabbit IgG conjugated with fluorescein isothiocyanate (FITC). For nuclear counterstaining, the cells were mounted in a mounting medium including 4′6-diamidino-2-phenylindole (DAPI; Abcam, Cambridge, MA, USA). Fluorescent images were collected and analyzed using a Zeiss Axioskop 50 fluorescence microscope (Carl Zeiss, Jena, Germany) equipped with a XIMEA with CCD-xiD camera (Ximea, Lakewood, CO, USA).

### 3.6. Measurement of Intracellular Level of ROS

Intracellular ROS level was measured by flow cytometry using the peroxide-sensitive fluorescent probe 2′7′-dichlorofluorescin diacetate (DCF–DA). Briefly, cells (3.5 × 10^6^ cells/well) were plated in 6-well culture plates and treated with each compound for 6 h. Cells were then incubated with DCF–DA (25 μM) in PBS at 37 °C for 30 min, washed twice with PBS, and detached by treatment with trypsin-ethylenediaminetetraacetic acid (EDTA). The detached cells were collected and resuspended in PBS, and the fluorescence intensity of cells was measured using flow cytometry (BD FACSVerse, BD-Biosciences, San Jose, CA, USA).

### 3.7. Statistical Analysis

The statistical significance of the differences among groups was analyzed using Student’s t-test. The results were expressed as means with a standard error of the mean (SEM) of 3 experiments for each group unless otherwise indicated, and a *p* value of less than 0.05 was considered statistically significant. All statistical tests were two-sided.

## Figures and Tables

**Figure 1 molecules-26-02316-f001:**
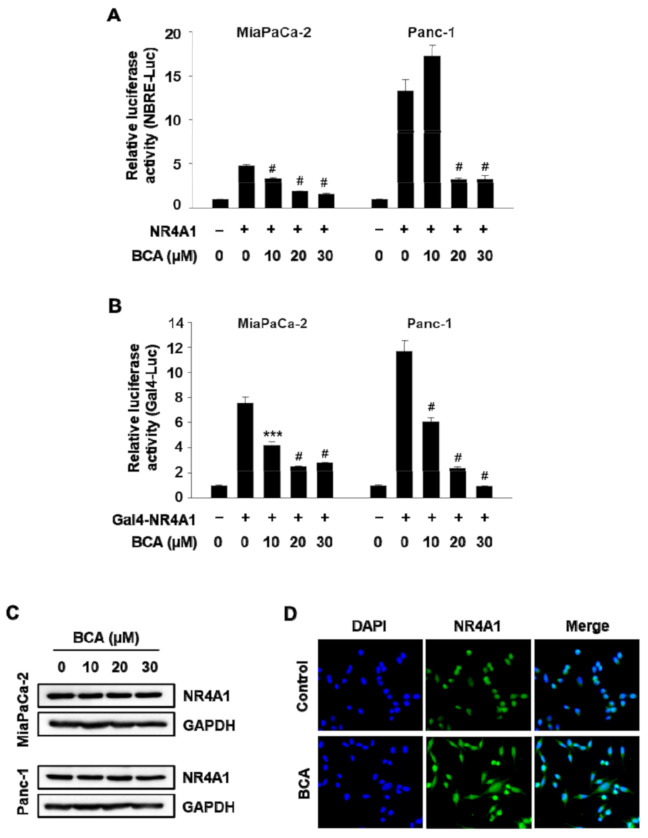
Broussochalcone A (BCA) inhibits transactivation of the orphan nuclear receptor 4A1 (NR4A1) in pancreatic cancer cells. (**A**) Cells were cotransfected with NBRE-Luc (25 ng) and Flag-NR4A1 (12.5 ng) for 5 h, and then treated with various concentrations of BCA for 18 h. ^#^
*p* < 0.001 vs. dimethyl sulfoxide (DMSO) + NR4A1. (**B**) Cells were cotransfected with Gal4-RE-Luc (25 ng) and 5 ng of Gal4-NR4A1 for 5 h, and then treated with BCA for 18 h. Luciferase activity (relative to β-galactosidase) was determined, and the corresponding empty vector was used as a control. The results are presented as means ± SEM (standard error of the mean). *** *p* < 0.005 and ^#^
*p* < 0.001 vs. DMSO + Gal4-NR4A1. (**C**) Cells were treated with BCA for 24 h, and whole cell lysates were analyzed by Western blot analysis. Glyceraldehyde 3-phosphate dehydrogenase (GAPDH) was used as a loading control. (**D**) Subcellular localization of NR4A1. MiaPaCa-2 cells were treated with either DMSO or BCA 15 μM for 3 h, and endogenous NR4A1 was stained and visualized as described in the Section 3.

**Figure 2 molecules-26-02316-f002:**
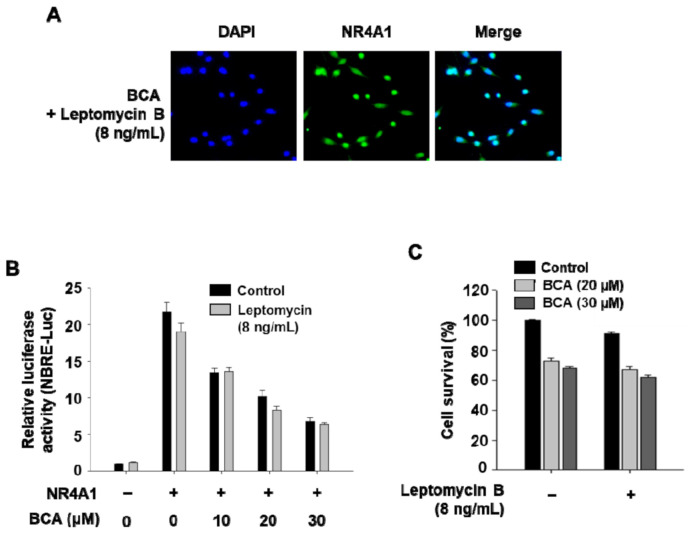
Blockage of the nuclear export of NR4A1 has no effect on BCA-mediated inhibition of NR4A1 transactivation and cell growth in pancreatic cancer cells. (**A**) Subcellular localization of NR4A1. MiaPaCa-2 cells were treated with BCA (20 μM) in the presence or absence of leptomycin B for 3 h, and endogenous NR4A1 was stained and visualized as described in the Section 3. (**B**) Cells were transfected with NBRE-Luc (25 ng) for 5 h and then treated with BCA in the presence or absence of leptomycin B for 18 h. Luciferase activity (relative to β-galactosidase) was determined, and the corresponding empty vector was used as a control. (**C**) Cells were treated with BCA in the presence or absence of leptomycin B for 24 h, and cell viability was determined as described in the Section 3. The results are presented as means ± SEM. No significant difference was observed between BCA alone and BCA + leptomycin B.

**Figure 3 molecules-26-02316-f003:**
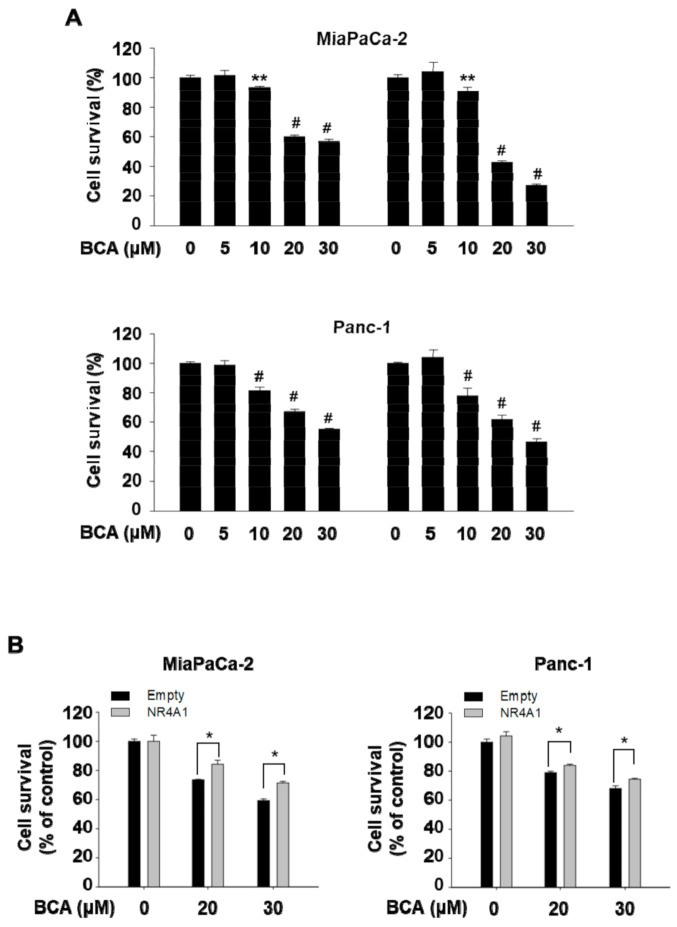
BCA inhibits cell proliferation in pancreatic cancer cells. (**A**) Cells were treated with various concentrations of BCA for indicated time, and cell viability was determined as described in the Section 3. (**B**) Cells were transfected with either Flag-empty or Flag-NR4A1 for 6 h, and at 24 h after transfection, cells were treated with BCA for 24 h. The results are presented as means ± SEM. * *p* < 0.05, ** *p* < 0.01, and ^#^
*p* < 0.001 vs. DMSO.

**Figure 4 molecules-26-02316-f004:**
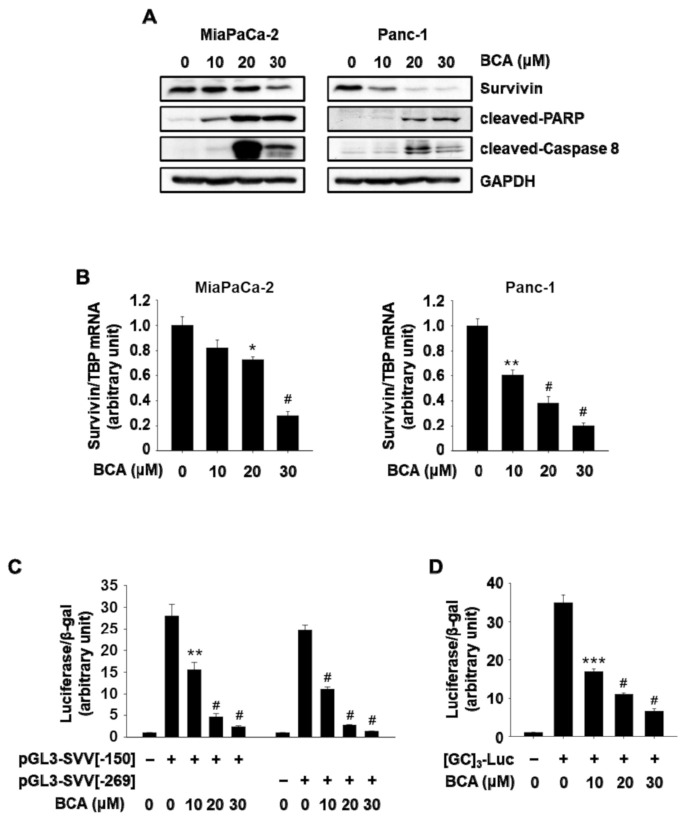
BCA inhibits Sp1-mediated survivin expression and induces apoptosis in pancreatic cancer cells. (**A**) Cells were treated with BCA for 24 h, and whole cell lysates were analyzed by Western blot analysis. GAPDH was used as a loading control. (**B**) Cells were treated with BCA for 18 h, and survivin mRNA level was determined by quantitative real-time PCR as described in the Section 3. (**C**) Panc-1 cells were transfected with 25 ng of pGL3-SVV(−150) or pGL3-SVV(−269) for 5 h and then treated with BCA for 18 h. (**D**) Panc-1 cells were transfected with 25 ng of [GC]_3_-Luc for 5 h and then treated with BCA for 18 h. Luciferase activity (relative to β-galactosidase) was determined, and the corresponding empty vector was used as a control. The results are presented as means ± SEM. * *p* < 0.05, ** *p* < 0.01, *** *p* < 0.005, and ^#^
*p* < 0.001 vs. DMSO.

**Figure 5 molecules-26-02316-f005:**
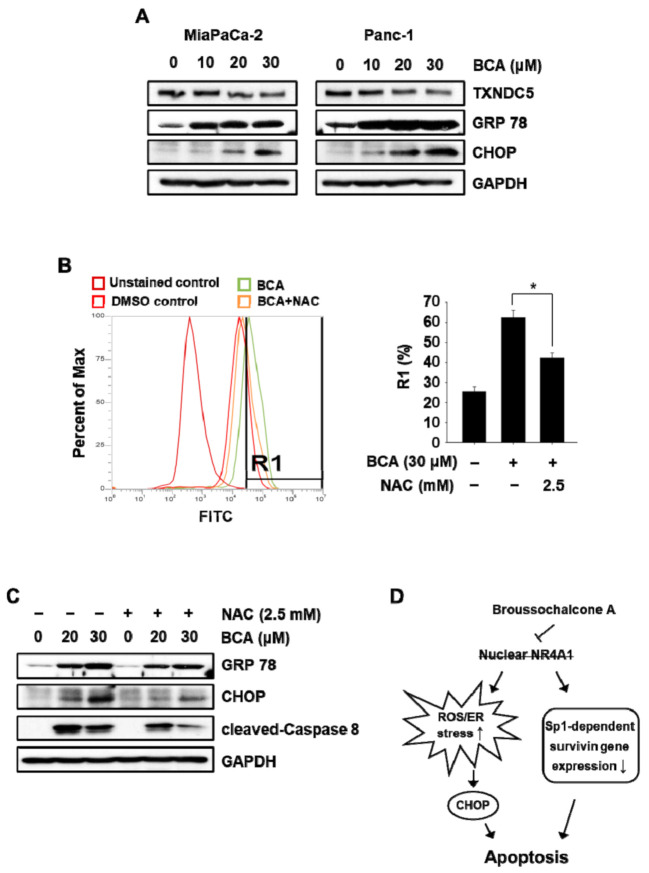
BCA induces ROS-mediated ER stress in pancreatic cancer cells. (**A**) Cells were treated with BCA for 24 h, and whole-cell lysates were analyzed by Western blot analysis. GAPDH was used as a loading control. (**B**) MiaPaCa-2 cells were treated with BCA for 6 h, and intracellular ROS level was measured by flow cytometry as described in the Section 3. (**C**) MiaPaCa-2 cells were treated with BCA for 24 h in the presence or absence of N-acetylcysteine (NAC), and whole cell lysates were analyzed by Western blot analysis. GAPDH was used as a loading control. (**D**) Schematic diagram illustrating NR4A1-dependent apoptosis by BCA. * *p* < 0.05 vs. DMSO.

## Data Availability

The data presented in this study are available in insert article or Appendix A here.

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
