# Peer review of "Broussochalcone A Is a Novel Inhibitor of the Orphan Nuclear Receptor NR4A1 and Induces Apoptosis in Pancreatic Cancer Cells"

_molecules, 2021, doi:10.3390/molecules26082316_

Round 1

Reviewer 1 Report

Manuscript ID: molecules-1183092

“Broussochalcone A is a novel inhibitor of the orphan nuclear receptor NR4A1 and induces apoptosis in pancreatic cancer cells” submitted by authors Syng-Ook Lee et al discusses the potential of broussochalcone A (BCA) for therapy of pancreatic tumors that overexpress NR4A1 in-vitro. BCA partly inhibited the cell growth by inducing NR4A1-mediated apoptotic pathways in human pancreatic cancer cells. Further BCA downregulated specificity protein 1 (Sp1)-mediated expression of the anti-apoptotic protein, survivin, and induced endoplasmic reticulum (ER) stress-mediated apoptotic pathway.

The paper is current, well structured, in which the authors have clearly introduced the work and showcased the importance of broussochalcone A (BCA) for anticancer drug discovery. The isolation methods, & structural characterization are well documented. Overall, within context of natural product exploration for anticancer drug discovery this manuscript is current and the efforts of authors are commended. Therefore, this paper merits acceptance in Molecules after minor corrections.

General comment-

In the isolation section it is mentioned lines 71-74 “Among them, subfraction 10-3 (530 mg) was further purified with C18 reversed-phase column chromatography (ODS-A, YMC, Kyoto, Japan) by step-gradient elution with each of the different percentage of acetonitrile  (40–100%, 100 mL).” and later at line 75-77“The compound (800 mg) was identified as BCA by using different spectroscopic techniques, including UV absorption, 1D NMR (1H, 13C NMR), 2D NMR (1H-1H-COSY, HSQC, HMBC), and electron ionization-mass spectrometry (EI-MS). Was 800 mg isolated from fraction 10-3?? Please clarify. What is the other solvent used for C18 gradient elution along with acetonitrile?

HPLC method and/or solvent system used should be mentioned in supplementary information.

Line 78-79: The NMR used is mentioned as Jeol ECA-500 MHz NMR instrument (Jeol, Tokyo, Japan) operating at 400 MHz; while in description of 1H NMR data it is mentioned 500 MHz. please correct the frequency.

Melting point of the yellow BCA powder should be recorded.

The carefully designed experiments with overexpression of NR4A1 partly rescued the growth inhibition by BCA (@ 20 and 30 µM) in both MiaPaCa-2 and Panc-1 cells as shown in Fig 3B. This enabled the authors to ascertain the mechanism of BCA-mediated growth inhibition of human pancreatic cancer cells via inactivation of NR4A1.

Without the in-vivo efficacy of BCA or clinical assessment data the written claim in the last part of abstract “potential treatment option for patients with pancreatic tumors that overexpress NR4A1” seems farfetched. Please revise statement accordingly.

Please insert DOI for all the references in the references section.

Author Response

  1. In the isolation section it is mentioned lines 71-74 “Among them, subfraction 10-3 (530 mg) was further purified with C18 reversed-phase column chromatography (ODS-A, YMC, Kyoto, Japan) by step-gradient elution with each of the different percentage of acetonitrile  (40–100%, 100 mL).” and later at line 75-77“The compound (800 mg) was identified as BCA by using different spectroscopic techniques, including UV absorption, 1D NMR (1H, 13C NMR), 2D NMR (1H-1H-COSY, HSQC, HMBC), and electron ionization-mass spectrometry (EI-MS). Was 800 mg isolated from fraction 10-3?? Please clarify. What is the other solvent used for C18 gradient elution along with acetonitrile?     
        There was an error in the method description. The amount of subfraction 10-3 was 9 g, and so we have now corrected it in the revised manuscript (Section 2.1.). And the solvent used for C18 gradient elution along with acetonitrile was distilled water, and we have now added this information in new supplementary Table 1.

  2.  HPLC method and/or solvent system used should be mentioned in supplementary information.      
    As suggested, the HPLC method has been added in the supplementary information (new Supplementary Table 1). 
  3.  “Line 78-79: The NMR used is mentioned as Jeol ECA-500 MHz NMR instrument (Jeol, Tokyo, Japan) operating at 400 MHz; while in description of 1H NMR data it is mentioned 500 MHz. please correct the frequency.        
    There was an error in the method description. 500 MHz is correct, and so we have now corrected it in the revised manuscript (Section 2.1.). 
  4.  “Melting point of the yellow BCA powder should be recorded.     
        Melting point of the yellow BCA powder was 183-184℃, and this information has now been added in the revised manuscript (Section 2.1.).
  5.  “….. Without the in-vivo efficacy of BCA or clinical assessment data the written claim in the last part of abstract “potential treatment option for patients with pancreatic tumors that overexpress NR4A1” seems farfetched. Please revise statement accordingly.       
      As suggested, the sentence has been revised (Abstract section).
  6.  “Please insert DOI for all the references in the references section       
      As suggested, DOI for all the references has been added in the revised manuscript.

Reviewer 2 Report

Lee et al., studied the “Broussochalcone A is a novel inhibitor of the orphan nuclear receptor NR4A1 and induces apoptosis in pancreatic cancer cells”. The authors have well planned the study and executed the methods.

In general, chalcones, either synthetic or natural, have broad anti-cancer properties. A piece of brief information regarding chalcones against various cancers should be presented in the introduction. This reference would be helpful https://doi.org/10.2174/0929867325666180530094120

In the introduction, the importance of natural products against pancreatic cancer should be presented. This reference would be helpful https://doi.org/10.1016/j.apsb.2019.11.008

Line 62: It is essential to identify the exact location from where the B. papyrifera harvested, although it is purchased from the market. Because, as the authors know, chemical constituents composition may change based on the region harvesting.

Section 2: Bring all procedures related to biological assays into the main manuscript.

Figure 2: Though these are interesting, microscopy study alone doesn’t support the data in a strong manner. Reliably, I would recommend authors separate the nuclear and cytosol fractions and estimate the NR4A1. However, in future studies, this experiment would be useful.

Figure 4, 5: Change c-PARP, c-Caspase 8 to cleaved-PARP, cleaved-Caspase-8

Though authors have cited their lab work many times, they are reasonable and mandatory.

Reference 9 is missing, please take care.

Author Response

1. “In general, chalcones, either synthetic or natural, have broad anti-cancer properties. A piece of brief information regarding chalcones against various cancers should be presented in the introduction. This reference would be helpful https://doi.org/10.2174/0929867325666180530094120      

  As suggested, a piece of brief information regarding anticancer properties of chalcones has now been added in the revised manuscript (the Introduction section).  

2. “In the introduction, the importance of natural products against pancreatic cancer should be presented. This reference would be helpful https://doi.org/10.1016/j.apsb.2019.11.008     

    As suggested, the importance of natural products against pancreatic cancer has been presented in the revised manuscript (the Introduction section). 

3.  “Line 62: It is essential to identify the exact location from where the B. papyrifera harvested, although it is purchased from the market. Because, as the authors know, chemical constituents composition may change based on the region harvesting.     

    As suggested, the location from where the B. papyrifera harvested has now been added in the revised manuscript (Section 2.1.). 

4. “Section 2: Bring all procedures related to biological assays into the main manuscript.    

     As suggested, all procedures related to biological assays have been moved to the main manuscript. 

5. “Figure 2: Though these are interesting, microscopy study alone doesn’t support the data in a strong manner. Reliably, I would recommend authors separate the nuclear and cytosol fractions and estimate the NR4A1. However, in future studies, this experiment would be useful.     

  As recommended, in our future studies, we will further confirm the presence of NR4A1 after separating the nuclear and cytosol fractions. 

6 . “Figure 4, 5: Change c-PARP, c-Caspase 8 to cleaved-PARP, cleaved-Caspase-8        

 As suggested, we have changed c- to cleaved- in Figs. 4 and 5. 

7. “Reference 9 is missing         

  There seem to have an error while converting the manuscript to form for peer review, and the original manuscript was not missing references (the total number of references was 18).
